# Postoperative acute respiratory dysfunction and the influence of antibiotics after acute type A aortic dissection surgery: A retrospective analysis

**Christina M. Möller**[1], **Peter-Paul Ellmauer**[1]**, Florian Zeman**[2]**, Diane Bitzinger**[1]**, Bernhard Flörchinger**[3]**, Bernhard M. Graf**[1]**, York A. Zausig**[1,4]*

1 Department of Anesthesiology, University Hospital Regensburg, Regensburg, Bavaria, Germany, 2 Center of Clinical Studies, University Hospital Regensburg, Regensburg, Bavaria, Germany, 3 Department of Cardiac Surgery, University Hospital Regensburg, Regensburg, Bavaria, Germany, 4 Department of Anesthesiology and Operative Intensive Care Medicine, Aschaffenburg-Alzenau Hospital, Aschaffenburg, Bavaria, Germany

* York.zausig@ukr.de, York.zausig@klinikum-ab-alz.de

**Data Availability Statement:** These data are legally restricted and cannot be shared publicly. Due to the new General Data Protection Regulation (GDPR;

## Abstract

### Objectives

Surgery for acute type A aortic dissection is associated with several perioperative complications, such as acute respiratory dysfunction (ARD). The aim of this study was to investigate perioperative risk factors involved in the development of ARD and whether antibiotic treatment has an impact.

### Methods

243 patients underwent surgery for acute type A aortic dissection between 2008 and 2017. The patients were retrospectively divided into the ARD and NON-ARD group. ARD was defined as $PaO_2/FiO_2 \leq 200$ mmHg (PF ratio) within 48 hours after surgery. All patients received either narrow- or broad-spectrum antibiotics.

### Results

After the exclusion of 42 patients, 201 patients were analyzed. The PF ratio of the ARD group was significantly lower than of the NON-ARD group within the first 7 days. ARD patients (n = 111) were significantly older ($p = .031$) and had a higher body mass index (BMI) ($p = .017$). ARD patients required longer postoperative ventilation (2493 vs. 4695 [min], $p = .006$) and spent more days in the intensive care unit (7.0 vs. 8.9 [days], $p = .043$) compared to NON-ARD. The mortality was significantly lower for ARD than for NON-ARD patients ($p = .030$). The incidence of pneumonia was independent of the antibiotic treatment regime ($p = .391$). Renal and neurological complication rate was higher in patients treated with broad-spectrum antibiotic.

EU 2016/679) by the European Union Government, our data protection commissioner recommends that the data from this study, containing potentially identifying or sensitive patient information, are only available upon request. Data are stored on the institutional server at the Department of Anesthesiology at the University of Regensburg. Interested, qualified researchers may request the data by contacting Dr. Michael Gruber at michael.gruber@ukr.de.

**Funding:** The authors received no specific funding for this work.

**Competing interests:** The authors have declared that no competing interests exist.

## Conclusion

ARD is the main complication (55%) that occurs approximately 24 hours after surgery for acute type A aortic dissection. The preoperative risk factors for ARD were higher age and increased BMI. Patients on broad-spectrum antibiotics did not show an improved postoperative outcome compared to patients with narrow-spectrum antibiotics.

## Introduction

Acute type A aortic dissection is a life-threatening disease with an incidence of approximately 2–16 cases/100 000 inhabitants in Europe per year and a preoperative mortality of approximately 17.6% [1–3]. After the patient survived surgery, there is still a high in-hospital mortality of more than 25% [4]. The high mortality rate is caused by various perioperative complications [5, 6].

One of the major and most frequent complications is acute respiratory dysfunction (ARD). Approximately 13% of these patients suffer from ARD, which usually occurs within the first 72 hours after surgery [7]. The consequences of pulmonary dysfunction are a prolonged stay in the intensive care unit (ICU), longer time of mechanical ventilation, higher risk of pneumonia, higher hospital costs and an increased rate of in-hospital mortality [8]. Currently, neither the pathophysiology of ARD nor its risk factors are well understood.

This study aimed to outline perioperative risk factors associated with the development of postoperative ARD in patients suffering from acute type A dissection. Furthermore, we investigated whether the incidence and development of ARD is influenced by the postoperative early use of narrow-spectrum vs. broad-spectrum antibiotics.

## Materials and methods

After obtaining approval from the University of Regensburg's Ethics Commission (No: 16-104-0278), all patients who underwent surgery for acute type A aortic dissection at the University Hospital of Regensburg from November 2008 to April 2017 were retrospectively examined. Due to the design of the study an informed consent was not necessary. Over the first seven days after admission to the ICU, detailed information, e.g. monitoring of blood gas analysis (BGA) and laboratory parameters, was obtained. Anesthesia and surgery protocols, electronic patient files, particularly from the ICU, and physician's letters were included for data collection.

### Surgical management

The operation followed a standardized procedure [9]. After thoracotomy, the heart and aorta were exposed, and blood circulation was carried out with a cardiopulmonary bypass. Hypothermia was induced for cerebral protection. The management of hypothermia did not change over the time. Depending on the length of the dissection, the ascending aorta, ascending aorta with the proximal hemiarch or ascending aorta and the total arch were replaced with a vascular prosthesis. If necessary, the aortic valve was either reconstructed or replaced [10].

### Anesthetic and ICU management

Midazolam and/or propofol in combination with sufentanil and pancuronium were mainly used for anesthetic induction. For the maintenance of anesthesia, in general, continuous

sufentanil and sevoflurane were given. The ventilation goals were a tidal volume of 6–8 ml kg $^{-1}$, a respiratory rate of 10–12 min$^{-1}$ and a positive end-expiratory pressure (PEEP) of $\geq 5$ mmHg. Thiopental and/or cortisone were regularly given for cerebral protection during extracorporeal bypass. Patients with a hemoglobin concentration of $< 8.0$ g dl$^{-1}$ received blood transfusion.

In general, postoperative care included pressure-controlled protective ventilation, infusions and catecholamine regimes following standards [11]. The main criteria for extubation were adequate gas exchange, the level of consciousness (for example GCS and RASS), normothermia, hemodynamic stability, a positive cough reflex and no significant bleeding.

Patients received either narrow-spectrum or broad-spectrum antibiotics depending on the physician in charge. Narrow-spectrum antibiotic was normally a one day course of cefuroxime and broad-spectrum antibiotic was normally piperacillin/tazobactam for several days. Both were started within the first 24 hours after admission.

Further individual treatment regimes, including positioning of the patient and extracorporeal membrane oxygenation (ECMO) therapy, was also dependent on the physician in charge.

## ARD definition

Currently, there is no standardized definition for ARD after acute type A aortic dissection surgery [5, 7, 8, 12]. Therefore, we used the latest Berlin definition for moderate acute respiratory distress syndrome (ARDS) and defined ARD as PaO$_2$/FiO$_2$ (PF ratio) $\leq 200$ mmHg in at least four out of six BGA within 48 hours after admission to the ICU [13].

Patients transferred to the intermediate care (IMC) or general ward $< 48$ hours after admission on the ICU were considered to have a PF ratio $> 200$ mmHg if the patient did not return to the ICU and/or died. For extubated patients, we used a conversion table to determine the FiO$_2$ [14]. ARD patients with postoperative pleural effusion, hemo-/pneumothorax or pneumonia were not excluded. Patients without ARD were considered NON-ARD.

## Statistical analysis

The statistical analysis was performed in cooperation with the Department of Biometry at the Centre for Clinical Studies at the University Hospital Regensburg. The SPSS statistics program 23.0 (IBM Corp., Armonk, NY, USA) was used for statistical analysis. Pre-, intra- and postoperative variables are presented as mean ± standard deviation for normally distributed variables, as the median (interquartile range) for variables with a skewed distribution and as absolute and relative frequencies for categorical variables. Risk factors for the development of ARD in association with the pre- and intraoperative variables were analyzed by using univariable logistic regression models. The odds ratio (OR) and the corresponding 95% confidence interval (95% CI) are presented as effect estimates. Differences in the postoperative variables between ARD and NON-ARD patients as well as between patients who received narrow- and broad-spectrum antibiotics were analyzed by using a Student's $t$-test for normally distributed variables, the Mann-Whitney $U$ test for non-normally distributed variables and the $\chi^2$ test of independence for categorical data. A $p$-value $< 0.05$ was considered statistically significant. No adjustments for multiple testing were made due to the exploratory nature of the study.

## Results

Overall, 243 cases were analyzed retrospectively. Forty-two patients were excluded due to death $\leq 48$ hours after surgery (n = 15), ECMO therapy $\leq 48$ hours after surgery (n = 8), massive intraoperative transfusions ($\geq 20$ fresh frozen plasma (FFP) and/or $\geq 10$ red blood cells

(RBCs)) (n = 6) or insufficient documentation of BGAs within the first 48 hours (n = 13). A total of 201 patients were included in this study (Fig 1).

A total of 111 out of 201 (55%) patients met our criteria for ARD, which was the most frequent postoperative complication. The incidence of postoperative respiratory failure changed from year to year; however, a trend along the study period could not be detected. Fig 2 shows the mean PF ratio in the respective groups during the time of observation. Each PF ratio between the ARD and NON-ARD groups was significantly different from the preoperative admission until the seventh postoperative day. The lowest PF ratio occurred within the first 24 hours after surgery. Age (OR = 1.03, $p$ = .031) and body mass index (BMI) (OR = 1.09, $p$ = .017) were found to be significant risk factors for the occurrence of an ARD. Leukocytes, creatine kinase (CK) and other clinical parameters showed no significant predictive value (Table 1). No parameters involved in the intraoperative management revealed a predictive value for the development of ARD (Table 2).

The postoperative variables are presented in Table 3. When compared to NON-ARD patients, ARD patients had a longer median ventilation time (NON-ARD: 2493 (912–6922) [min] vs. ARD: 4695 (1852–8918) [min], $p$ = .006) and a longer ICU stay (NON-ARD: 7.0 ± 6.5 [days] vs. ARD: 8.9 ± 6.5 [days], $p$ = .043). The mortality of ARD patients was lower than the mortality of NON-ARD patients (NON-ARD: 10% (9/90) vs. ARD: 3% (3/111), $p$ = .030). In the NON-ARD group, three out of nine patients died due to cardiac complications, and six patients died due to cerebral complications. In the ARD group, two patients died from cardiac complications and one patient died from cerebral complications. Postoperative complications, such as pericardial effusion or tamponade, atrial fibrillation, hemo-/pneumothorax, pleural effusion, renal and neurological complication, were evenly distributed in both groups. There were a few more cases of pneumonia and tracheotomy in the ARD group than in the

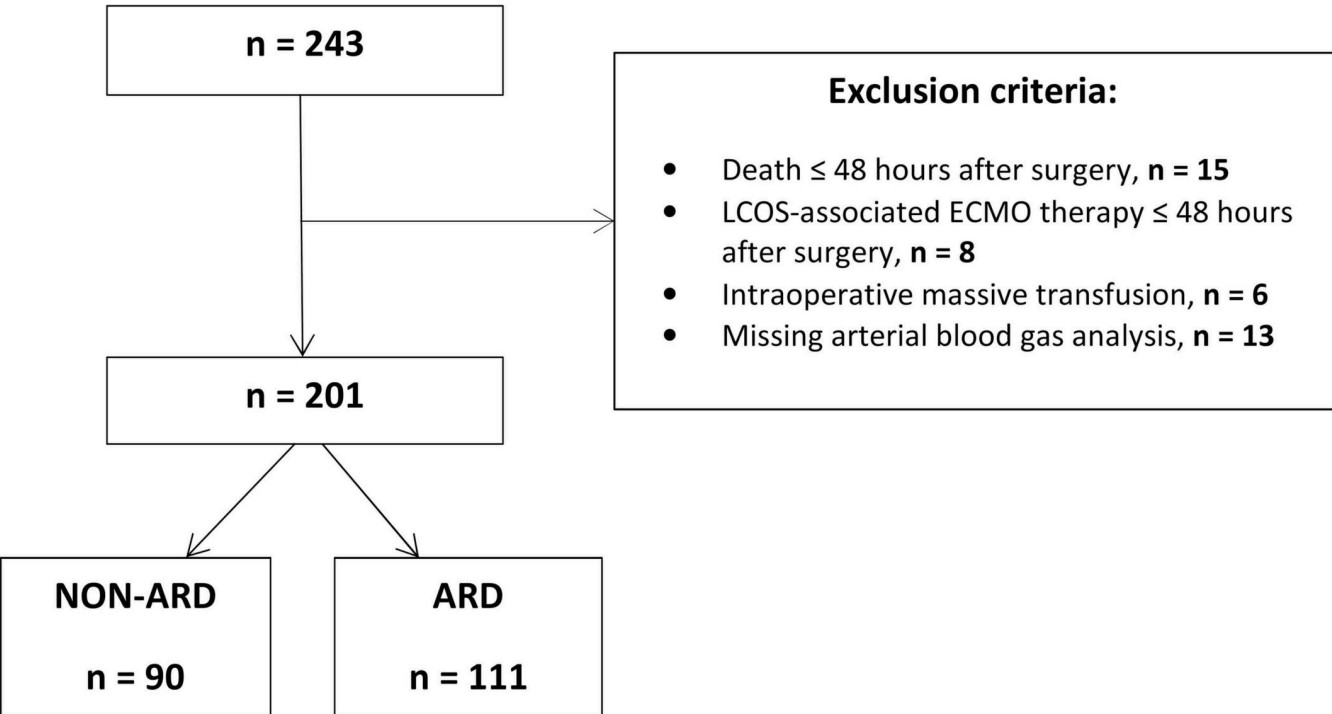

**Fig 1. Trial profile.** Flow chart showing the number of excluded patients, the causes of exclusion and the group selection. ARD: acute respiratory dysfunction; ECMO: extracorporeal membrane oxygenation; LCOS: low cardiac output syndrome.

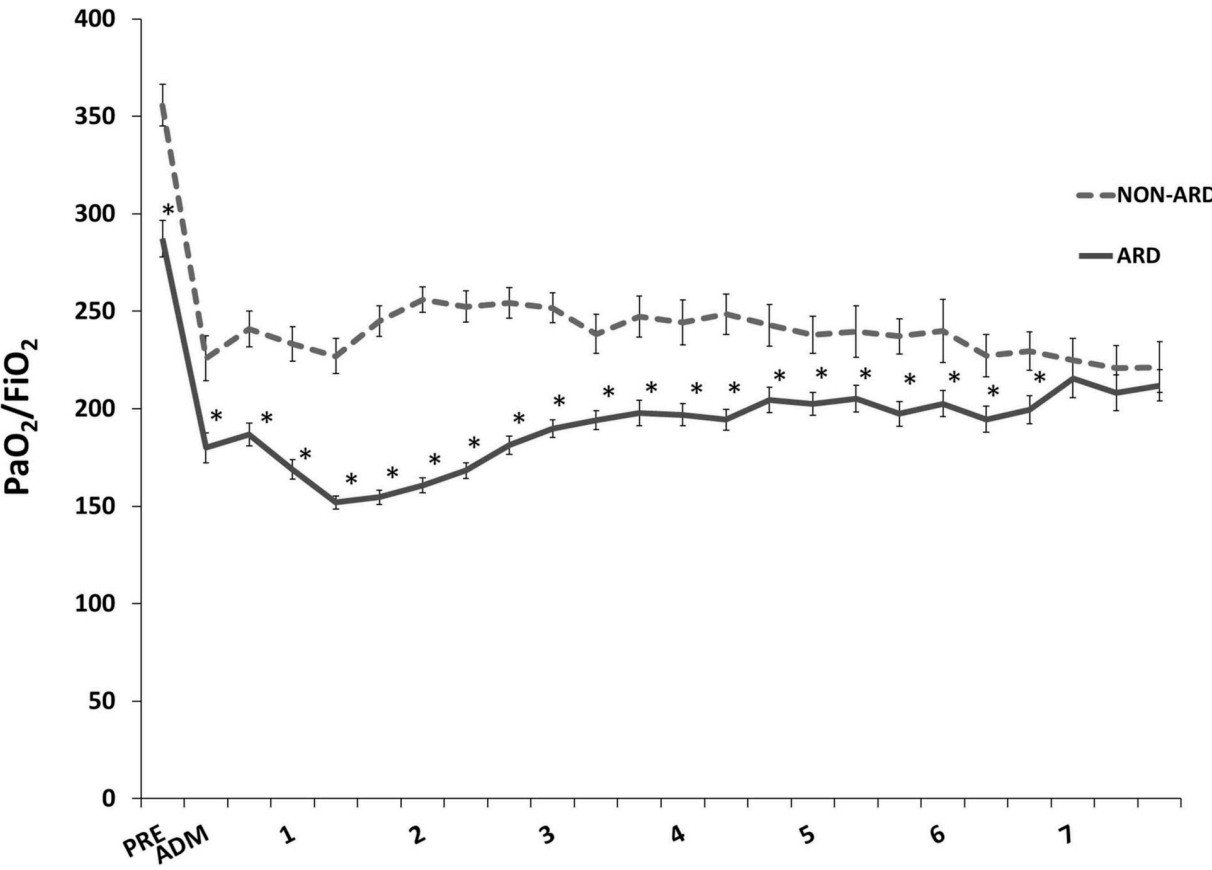

**Fig 2. PF ratio throughout hospital stay.** PF ratio [mmHg] is presented as the mean value and standard error of the two groups. ADM: ICU admission; ARD: acute respiratory dysfunction; PRE: preoperative. * ≙ p < 0.05; NON-ARD vs. ARD at day 1–7: at 8, 16 and 24 hours.

NON-ARD group. The blood lactate levels measured on the first postoperative day were significantly higher in the ARD group than in the NON-ARD group ($p = .001$; Table 3).

All patients were given antibiotics. A total of 42% of the patients (n = 84) received a narrow-spectrum antibiotic and 58% (n = 117) received a broad-spectrum antibiotic (Fig 3). In the NON-ARD group 43% of the patients (n = 39) were treated with a narrow-spectrum antibiotic and 57% (n = 51) were treated with a broad-spectrum antibiotic. In the ARD group 41% (n = 45) and 59% (n = 66) were treated with a narrow- or a broad-spectrum antibiotic, respectively. Pneumonia cases were equally diagnosed in both groups independently of the applied antibiotic medication ($p = .391$). In both groups the incidence of postoperative complication was tendentially higher in patients treated with a broad-spectrum antibiotic (Fig 3). In the NON-ARD group neurological complication were significantly more often, while in the ARD group renal complication were significantly more often when treated with a broad-spectrum antibiotic ($p < .001$ and $p = .039$). The hospital mortality was highest in the NON-ARD group treated with a broad-spectrum antibiotic ($p = .009$).

Independently of the ARD and NON-ARD group, patients treated with broad-spectrum antibiotic spent two additional days in the ICU ($p = .061$) and had a longer median ventilation time (narrow-spectrum antibiotic: 1607 (808–8242) [min] vs. broad-spectrum antibiotic: 4695 (2098–7941) [min], $p = .002$) compared to patients treated with narrow-spectrum antibiotic. Of all patients that died during their hospital stay 10 out of the 12 patients belonged to the broad-spectrum antibiotic group ($p = .069$). The choice of antibiotics did not influence the

**Table 1. Preoperative variables.**

| Variable | NON-ARD | ARD | OR (95% CI) | *p*-value |
|---|---|---|---|---|
| | (n = 90) | (n = 111) | | |
| Mean age [years] | 58.6 ± 13.1 | 62.6 ± 12.1 | 1.03 (1.00, 1.05) | 0.031* |
| Male (ref. female) | 56 [43] | 75 [57] | 1.27 (0.71, 2.3) | 0.429 |
| BMI [kg/m$^2$] | 26.9 ± 3.9 | 28.4 ± 4.3 | 1.09 (1.02, 1.17) | 0.017* |
| ASA | 3.9 ± 0.4 | 3.9 ± 0.5 | 1.05 (0.59, 1.89) | 0.865 |
| Arterial hypertension (ref. no) | 67 [43] | 89 [57] | 1.39 (0.71, 2.70) | 0.333 |
| Coronary heart disease (ref. no) | 8 [42] | 11 [58] | 1.13 (0.43, 2.93) | 0.806 |
| Aneurysm (ref. yes) | 18 [51] | 17 [49] | 1.38 (0.67, 2.87) | 0.385 |
| AI (ref. yes) | 39 [46] | 45 [54] | 1.12 (0.64, 1.97) | 0.690 |
| Pericardial disease (ref. no) | 23 [42] | 32 [58] | 1.18 (0.63, 2.21) | 0.605 |
| Ischemia of the organ (ref. no) | 2 [29] | 5 [71] | 2.08 (0.39, 10.96) | 0.390 |
| Ischemia of the limb (ref. no) | 10 [38] | 16 [62] | 1.35 (0.58, 3.13) | 0.489 |
| COPD (ref. no) | 4 [36] | 7 [64] | 1.45 (0.41, 5.11) | 0.566 |
| Nicotine (ref. no) | 17 [38] | 28 [62] | 1.45 (0.73, 2.86) | 0.285 |
| Diabetes mellitus (ref. no) | 4 [29] | 10 [71] | 2.13 (0.65, 7.03) | 0.215 |
| Chronic renal insufficiency (ref. yes) | 5 [46] | 6 [55] | 1.03 (0.30, 3.49) | 0.963 |
| Neurological complication (ref. yes) | 25 [46] | 29 [54] | 1.09 (0.58, 2.03) | 0.793 |
| CRP [mg/L] | 5.6 (2.9–41.8) | 6.6 (2.9–20.5) | 1.00 (0.99, 1.01) | 0.753 |
| Leukocytes [/nL] | 12.7 ± 4.0 | 12.7 ± 4.2 | 1.00 (0.93, 1.07) | 0.995 |
| CK [U/L] | 104.0 (69.0–169.0) | 100.0 (70.0–169.0) | 1.00 (1.00, 1.00) | 0.368 |

AI: aortic insufficiency; ARD: acute respiratory dysfunction; ASA: Score of the American Society of Anesthesiologists; BMI: body mass index; CK: creatine kinase; COPD: chronic obstructive pulmonary disease; CRP: C-reactive protein; Ischemia of the organ: heart, liver or kidney; Neurological complication: includes hemiplegia, paresis, seizure or stroke; Pericardial disease: includes effusion or tamponade.

All data are presented as the mean ± standard deviation, number [row-wise %] or median (IQR = interquartile range).

*: $p < 0.05$; *p*-value: logistic regression. Odds ratio (OR) and 95% confidence interval (95% CI).

incidence of pneumonia (narrow-spectrum antibiotic: 13% vs. broad-spectrum antibiotic: 14%, $p$ = .905).

## Discussion

This retrospective study shows that ARD is the main complication that occurred within 24 hours of acute type A aortic dissection surgery at a university hospital in Germany. The preoperative risk factors for ARD are a higher age and an increased BMI. ARD patients required a longer time of ventilation and an extended stay in the ICU, tended to develop pneumonia more often and needed tracheotomy more frequently than NON-ARD patients. Early broad-spectrum antibiotics did not influence the incidence of pneumonia and did not seem to be advantageous in preventing the development of ARD.

Postoperative ARD is a typical complication after non-cardiac and cardiac surgery [15, 16]. The incidence of ARD after surgery for acute type A aortic dissection is 8–48.5% [5, 7, 8, 12]. The respective studies have usually defined ARD on the basis of the Berlin definition for ARDS [13]. However, the PF ratio and time of measurement differed among all studies. For example, Girdauskas et al. defined ARD as the reduction of the PF ratio < 150 mmHg within 72 hours after surgery. These authors found an ARD incidence of 13% [7]. Chen et al., who defined ARD as a PF ratio < 300 mmHg within 7 days after surgery, reported an incidence of 12.7% [8]. In the present study, the incidence of ARD was higher (55%) than that in previous studies. Obviously, the chosen PF ratio of ≤ 200 mmHg within the observation period of 48

**Table 2. Intraoperative variables.**

| Variable | NON-ARD | ARD | OR (95% CI) | p-value |
|---|---|---|---|---|
| | (n = 90) | (n = 111) | | |
| Surgical procedure: | | | | |
| Aortic root replacement | 51 [50] | 51 [50] | reference | |
| Hemiarch replacement | 29 [39] | 45 [61] | 1.55 (0.85, 2.85) | 0.156 |
| Total arch replacement | 10 [42] | 14 [58] | 1.40 (0.57, 3.44) | 0.463 |
| Duration of operation [min] | 298 ± 64 | 297 ± 78 | 1.00 (1.00, 1.00) | 0.945 |
| CPB time [min] | 172 ± 50 | 172 ± 58 | 1.00 (1.00, 1.01) | 0.987 |
| Cross-clamp time [min] | 96 ± 34 | 99 ± 32 | 1.00 (0.99, 1.01) | 0.620 |
| Circulatory arrest time [min] | 37 ± 18 | 40 ± 21 | 1.01 (0.99, 1.03) | 0.395 |
| Cerebral perfusion time [min] | 32 ± 21 | 37 ± 22 | 1.01 (0.99, 1.03) | 0.228 |
| Reperfusion time [min] | 61 ± 23 | 56 ± 20 | 1.01 (0.99, 1.03) | 0.282 |
| Hypothermia: | | | | |
| deep ≤ 20°C | 11 [58] | 8 [42] | reference | |
| moderate 20.1–28°C | 61 [48] | 66 [52] | 1.49 (0.56, 3.94) | 0.425 |
| mild 28.1–34°C | 5 [56] | 4 [44] | 1.10 (0.22, 5.45) | 0.907 |
| Thiopental (ref. yes) | 70 [46] | 82 [54] | 1.30 (0.67, 2.52) | 0.432 |
| Cortisone (ref. no) | 26 [41] | 37 [59] | 1.23 (0.67, 2.25) | 0.500 |
| RBC [units] | 2.7 ± 2.0 | 3.1 ± 2.1 | 1.08 (0.94, 1.24) | 0.287 |
| FFP [units] | 6.1 ± 3.3 | 6.7 ± 3.9 | 1.05 (0.97, 1.14) | 0.217 |
| Platelets [units] | 2.4 ± 0.9 | 2.5 ± 1.0 | 1.06 (0.78, 1.43) | 0.713 |
| PCC [units] | 3658 ± 1834 | 3645 ± 1813 | 1.00 (1.00, 1.00) | 0.959 |

ARD: acute respiratory dysfunction; Cortisone: e.g. prednisolone, dexamethasone; CPB time: cardiopulmonary bypass time; FFP: fresh frozen plasma; PCC: prothrombin complex concentrate; RBC: red blood cells.

All data are presented as the mean ± standard deviation or number [%]. *: $p < 0.05$; p-value: logistic regression. Odds ratio (OR) and 95% confidence interval (95% CI).

hours after surgery might be the reason that we observed a higher incidence of ARD. Furthermore, the selected inclusion and exclusion criteria might have had a substantial influence on the study results. For example, in the present study, extubated patients were included by using a conversion table to determine the $FiO_2$ for nasal probe and mask ventilation [14]. It is not clear how other studies addressed this issue. Moreover, only patients with early ECMO therapy, massive intraoperative transfusions, early death and insufficient documentation of BGAs were excluded in this study. In contrast, Girdauskas et al. and Chen et al. also excluded patients with cardiogenic pulmonary edema, pneumonia, pulmonary embolism or hemo-/pneumothorax [7, 8]. Interestingly, studies with a comparable definition of ARD and no defined exclusion criteria showed a similar incidence of 48.5% for ARD [12]. Our ARD patients required a longer time of ventilation with an extended stay in the ICU, tended to develop pneumonia and underwent tracheotomy more often than NON-ARD patients. This finding is supported by other studies who had similar results [7, 8, 12].

This study demonstrated that higher age and an increased BMI are risk factors for the development of ARD after acute type A aortic dissection surgery. This result is supported by other studies [12, 17]. Normally, obesity is known as a protective factor against admission to the ICU [18]. The meta-analysis by Ni et al. demonstrated that patients with ARDS and obesity have a lower mortality than non-obese patients [19]. In the present study, ARD patients suffering from obesity died significantly less frequently than NON-ARD patients. But it should not be neglected that it could also be a selection bias due to the ARD definition. In addition, oxygenation impairment, prolonged circulatory arrest time, operative procedures and massive

**Table 3. Postoperative variables.**

| Variable | NON-ARD | ARD | p-value |
|---|---|---|---|
| | (n = 90) | (n = 111) | |
| CRP 1 day [mg/L] | 91.9 ± 53.8 | 78.9 ± 51.9 | 0.101 |
| CRP 2 day [mg/L] | 176.4 ± 62.0 | 167.1 ± 59.1 | 0.281 |
| Leukocytes 1 day [/nL] | 9.9 ± 3.3 | 10.2 ± 3.4 | 0.484 |
| Leukocytes 2 day [/nL] | 12.4 ± 4.4 | 12.1 ± 3.9 | 0.631 |
| CK 1 day [U/L] | 599 (339–1068) | 512 (333–1354) | 0.557 |
| CK 2 day [U/L] | 642 (374–1532) | 752 (478–1730) | 0.357 |
| Lactate 1 day [mg/dL] | 20.3 ± 13.5 | 30.1 ± 26.7 | 0.001* |
| Lactate 2 day [mg/dL] | 11.5 ± 5.0 | 13.3 ± 9.2 | 0.074 |
| Pericardial disease | 14 [16] | 22 [20] | 0.433 |
| Pneumonia | 8 [9] | 19 [17] | 0.089 |
| Hemo-/Pneumothorax | 4 [4] | 8 [7] | 0.411 |
| Pleural effusion | 35 [39] | 51 [46] | 0.315 |
| Tracheotomy | 5 [6] | 15 [14] | 0.061 |
| Ventilation [min] | 2493 (912–6922) | 4695 (1852–8918) | 0.006* |
| Renal complication | 17 [19] | 23 [21] | 0.746 |
| RRT | 13 [14] | 20 [18] | 0.496 |
| Neurological complication | 29 [32] | 40 [36] | 0.571 |
| Delirium | 19 [21] | 24 [22] | 0.930 |
| ICU stay [days] | 7.0 ± 6.5 | 8.9 ± 6.5 | 0.043* |
| Hospital mortality | 9 [10] | 3 [3] | 0.030* |

ARD: acute respiratory dysfunction; CK: creatine kinase; CRP: C-reactive protein; ICU stay: intensive care unit stay; Neurological complication: includes hemiplegia, paresis, seizure or stroke; Pericardial disease: effusion or tamponade; Renal complication: renal insufficiency or failure; RRT: renal replacement therapy.

All data are presented as the mean ± standard deviation, number [%] or median (IQR = interquartile range).

*: $p < 0.05$; p-value: χ2; except for CK and ventilation (Mann-Whitney $U$ test) and CRP, ICU stay, lactate, leukocytes and SAPS (Student's t-test).

blood transfusion are also risk factors for ARD [12, 16]. The latter was not addressed in our study since patients with intraoperative massive transfusions were excluded from the analysis. A preoperative malperfusion of one or more organs is also associated with the occurrence of ARD [7]. Strikingly, patients with ARD showed a significantly higher lactate concentration–an indirect indicator for malperfusion–on the first postoperative day compared to that in NON-ARD patients in the presented manuscript [20].

After cardiac surgery, especially after aortic surgery, the incidence of ARD is high. Pathophysiological data suggest that systemic inflammation could cause postoperative ARD and is possibly triggered or aggravated by microbial processes [21, 22]. Based on this assumption, treatment with broad-spectrum antibiotics is commonly used early in the ICU. The advantage of this approach has not been demonstrated in the literature at this moment. All patients in our study received an antibiotic treatment. Narrow-spectrum and broad-spectrum antibiotics were equally distributed in the NON-ARD and ARD group. There was no difference in the incidence of pneumonia between the two groups. Although the groups had comparable preoperative demographic and medical characteristics, patients who received a broad-spectrum antibiotic therapy showed a worse postoperative outcome with a higher incidence of renal and neurological complication and death than patients who received narrow-spectrum antibiotics. Piperacillin/tazobactam, which is generally used as a broad-spectrum antibiotic treatment, is not commonly associated with nephrotoxicity [23]. However, it was observed that vancomycin in combination with piperacillin/tazobactam increases the number of acute kidney injuries

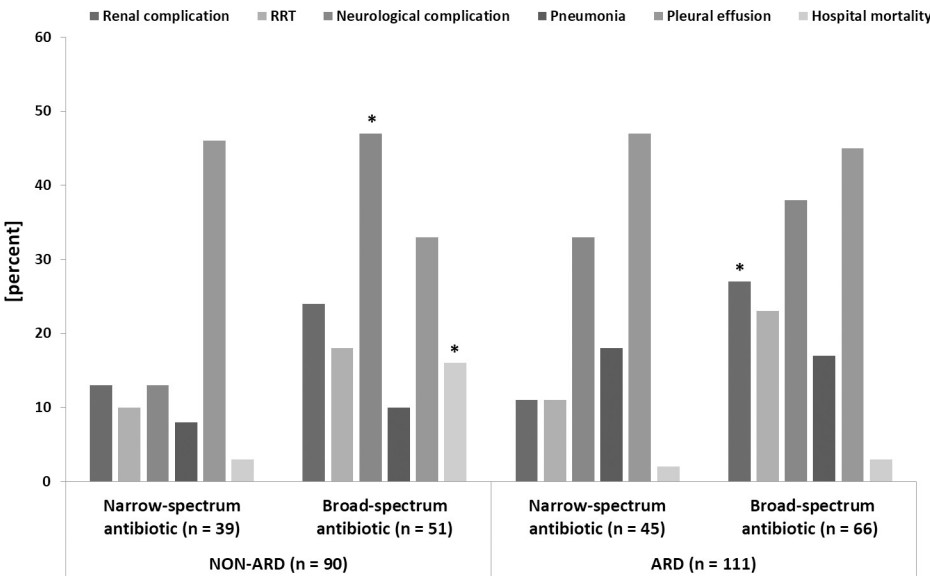

**Fig 3. Postoperative outcome and antibiotic treatment.** Comparison of postoperative complications between NON-ARD and ARD which are subdivided in the antibiotic groups. ARD: acute respiratory dysfunction; Neurological complication: includes hemiplegia, paresis, seizure or stroke; Renal complication: renal insufficiency or failure; RRT: renal replacement therapy. * ≙ p < 0.05 (Narrow-spectrum vs. broad-spectrum antibiotic in the NON-ARD and ARD group).

[24]. Among all of the patients, only 7 out of the 201 patients were treated with this antibiotic combination as an escalation of treatment during their hospital stay. Therefore, the side effects of broad-spectrum antibiotic treatment may only be a minor cause of the renal and neurological symptoms. However, other more potent factors that are not affected by the intensive care treatment might determine the patient outcome. First, an aortic dissection alone can cause kidney and brain malperfusion and damage prior to the surgical intervention or stay in the ICU [25]. However, we do not know which patients experienced malperfusion because this was not documented. Second, the surgical techniques and the cardiopulmonary bypass time associated with circulatory arrest may have led to organ damage [16]. Third, postoperative care with insufficient lung protective ventilation, without the prone position and without restrictive fluid-volume management may have aggravated postoperative respiratory dysfunction [21, 26]. Furthermore, some studies have described a kidney-lung crosstalk. It is assumed that the lungs and kidneys share common pathophysiologic pathways and could thereby damage each other [27]. Accordingly, ARD can affect renal insufficiency or contribute to it. It is also possible that a neurological disorder could have led to silent aspirations and contributed to ARD or pneumonia [28].

Currently, no studies on the prevention of ARD by prophylactic and therapeutic antibiotic treatment after surgery for type A aortic dissection exist. However, Mui et al. conducted a prospective, randomized study on perioperative antibiotic treatment after open appendectomy due to acute non-perforated appendicitis. Patients receiving a 5-day perioperative antibiotic treatment received no benefit compared to that in patients who received a single dose of preoperative antibiotics, whereas longer antibiotic treatment was associated with more postoperative complications [29]. Another randomized study found that in comatose survivors of out-of-hospital cardiac arrest, prophylactic antibiotics did not lead to a better outcome than a clinically-driven antibiotic treatment [30]. Similarly, patients in the present study who received a broad-spectrum, and presumably more effective, antibiotic therapy did not have better

postoperative outcomes and had more nephrological and neurological complications than patients who received a narrow-spectrum antibiotic treatment.

The present study has some potential limitations due to its retrospective design. First, our results are based on documented data and are therefore limited in their ability to be compared with those from other studies. Second, the different ARD definitions used in other studies make it difficult to compare their results with each other and with our results; and it might have had an impact on our results. To validate our results and to enhance the power of the study, a multi-centre study would be necessary. Third, the postoperative antibiotic therapy in the present study did not follow a standardized procedure. Although it seems that broad-spectrum antibiotic treatment does not influence the incidence of ARD or the occurrence of pneumonia, due to the design of our study, we cannot rule out that the outcome of this patient group would have been worse without broad-spectrum antibiotic treatment. Fourth, we cannot clarify if the operation technique and cerebral protection has changed over the years. This might have had an impact on the outcome of the patients. Further prospective randomized studies that employ defined standards for antibiotic treatment will be necessary to replicate the effect of prophylactic antibiotic therapy on ARD.

In conclusion, ARD is a common complication in the first 24 hours after acute type A aortic dissection surgery. Higher age and increased BMI were associated with the occurrence of postoperative ARD in our study. Patients who received broad-spectrum antibiotics did not show a lower incidence of pneumonia than patients who received narrow-spectrum antibiotics. Furthermore, patients who received broad-spectrum antibiotics had more postoperative complications.

## Author Contributions

**Conceptualization:** York A. Zausig.

**Data curation:** Christina M. Möller, Peter-Paul Ellmauer.

**Formal analysis:** Christina M. Möller, Florian Zeman.

**Investigation:** Diane Bitzinger, Bernhard Flörchinger.

**Methodology:** Florian Zeman, York A. Zausig.

**Project administration:** Peter-Paul Ellmauer, York A. Zausig.

**Software:** Florian Zeman.

**Supervision:** Peter-Paul Ellmauer, Bernhard M. Graf, York A. Zausig.

**Validation:** Diane Bitzinger, Bernhard Flörchinger.

**Visualization:** Christina M. Möller, Florian Zeman.

**Writing – original draft:** Christina M. Möller.

**Writing – review & editing:** Peter-Paul Ellmauer, Diane Bitzinger, Bernhard Flörchinger, Bernhard M. Graf, York A. Zausig.

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
