## [Decision Letter · Decision Letter 0]

13 Jan 2020

PONE-D-19-32192

Postoperative acute respiratory dysfunction and the influence of antibiotics after acute type A aortic dissection surgery: a retrospective analysis

PLOS ONE

Dear Mrs. Möller,

Thank you for submitting your manuscript to PLOS ONE. After careful consideration, we feel that it has merit but does not fully meet PLOS ONE’s publication criteria as it currently stands. Therefore, we invite you to submit a revised version of the manuscript that addresses the points raised during the review process.

We would appreciate receiving your revised manuscript by Feb 27 2020 11:59PM. To enhance the reproducibility of your results, we recommend that if applicable you deposit your laboratory protocols in protocols.io, where a protocol can be assigned its own identifier (DOI) such that it can be cited independently in the future. For instructions see: http://journals.plos.org/plosone/s/submission-guidelines#loc-laboratory-protocols

We look forward to receiving your revised manuscript.

Kind regards,

Andrea Ballotta

Academic Editor

PLOS ONE

Additional Editor Comments (if provided):

On the basis of the comments of the two reviewers i invite you as authors to revise your manuscript following the concerns and the issues raised up by the two reviewers.

Journal Requirements:

Reviewers' comments:

Reviewer's Responses to Questions

**Comments to the Author**

1. Is the manuscript technically sound, and do the data support the conclusions?

Reviewer #1: Yes

Reviewer #2: Yes

2. Has the statistical analysis been performed appropriately and rigorously? 

Reviewer #1: Yes

Reviewer #2: Yes

3. Have the authors made all data underlying the findings in their manuscript fully available?

Reviewer #1: Yes

Reviewer #2: Yes

4. Is the manuscript presented in an intelligible fashion and written in standard English?

Reviewer #1: Yes

Reviewer #2: Yes

5. Review Comments to the Author

Reviewer #1: Tahnk you for your paper.

The paper is interesting because it consider the impact of our antibiotic on the outcome of our patients, despite that in my opinion I think the use of broad spectrum antibiotic is influence non only by the lung status but also from the clinical analysis.

Patients who are going clinically worst need to be covered by broad spectrum non only because the lung problem.

I don't understand how patients who develop ARF can die less then who don't, it might be the selection of ARF definition.

Reviewer #2: The Authors investigated the incidence, risk factors and impact on clinical outcomes of acute respiratory failure after surgery for acute type A aortic dissection. The retrospective cohort spans from 2008 to 2017.

I only have a few comments:

- did surgical techniques and cerebral protection techniques change during the study period?

- most patients were managed with moderate hypothermia. Did this change over time? If so, did this impact on transfusional requirements? Do the Authors believe that this had an impact on respiratory complications?

- did the incidence of postoperative respiratory failure change along the study period?

6. PLOS authors have the option to publish the peer review history of their article (what does this mean?). If published, this will include your full peer review and any attached files.

Reviewer #1: No

Reviewer #2: Yes: Dr. Fabio Sangalli, MD, FASE

---

## [Author Response · Author response to Decision Letter 0]

17 Feb 2020

Dear editor and reviewers,

We would like to thank the academic editor Mrs. Balotta and the reviewers for their extensive and constructive review of our manuscript. 

We have revised our paper and hope that our careful revision and our respective statements are satisfying. The following is a detailed response to the editors’ and reviewers’ comments. 

Editor:

Journal Requirements:

-> Changes have been made in line 180, 300, 306, 313, 328, 342, 361, 381, 400, and 401.

-> PACE has been used before the first submission.

a) If there are ethical or legal restrictions on sharing a de-identified data set, please explain them in detail (e.g., data contain potentially identifyingor sensitive patient information) and who has imposed them (e.g., an ethics committee). Please also provide contact information for a data access committee, ethics committee, or other institutional body to which data requests may be sent.

-> As we pointed out, the data are legally restricted and cannot be shared publicly. They are available any time upon request by contacting our Department at the University of Regensburg. The head of our research laboratory, Dr. Michael Gruber takes care of a safe data backup.

-> https://www.ukr.de/kliniken-institute/Anaesthesiologie/Klinikteam___Mitarbeiter/Forschungslabor/index.php

Reviewer #1: Tahnk you for your paper.

The paper is interesting because it consider the impact of our antibiotic on the outcome of our patients, despite that in my opinion I think the use of broad spectrumantibiotic is influence non only by the lung status but also from the clinical analysis.

Patients who are going clinically worst need to be covered by broad spectrum non only because the lung problem.

I don't understand how patients who develop ARF can die less then who don't, it might be the selection of ARF definition.

-> Yes you are right, we were also surprised by our results. There might be a selection bias due to the ARD definition. However, other factors might indepently influence the outcome. For example, a recent meta-analysis showed that the body mass index can predict the clinical outcomes for patients with acute respiratory distress syndrome. Ni et al. showed that obesity and morbid obesity was associated with a lower mortality in patients with ARDS. We found a higher BMI in our ARD group. Accordingly, this might have influenced our outcome with a lower mortality. With regard to your very important annotation, we added the following sentence in line 218: “But it should not be neglected that it could also be a selection bias due to the ARD definition.” Furthermore, we revised our sentence in line 274: “Second, the different ARD definitions used in other studies make it difficult to compare their results with each other and with our results; and it might have had an impact on our results.” 

Reviewer #2: The Authors investigated the incidence, risk factors and impact on clinical outcomes of acute respiratory failure after surgery for acute type A aortic dissection. The retrospective cohort spans from 2008 to 2017.

I only have a few comments:

- did surgical techniques and cerebral protection techniques change during the study period?

-> This is a very important annotation. It might be comprehensible that over the time period of 9 years a change in the surgical technique and/or cerebral protection might have occurred. Unfortunately, this was not well documented. 

-> As we totally agree, we therefore changed the manuscript (line 282): “Fourth, we cannot clarify if the operation technique and cerebral protection has changed over the years. This might have had an impact on the outcome of the patients.”

- most patients were managed with moderate hypothermia. Did this change over time? If so, did this impact on transfusional requirements? Do the Authors believe that this had an impact on respiratory complications?

-> The management of hypothermia did not change over the time. Therefore, transfusional requirements were not different. These results are listed in table 2. To respond to this important annotation, we changed our manuscript (line 66): “The management of hypothermia did not change over the time.”

- did the incidence of postoperative respiratoryfailure change along the study period?

-> This is a very interesting question! We have looked at our data. We found a different incidence of postoperative respiratory failure per year. However, we did not see a trend over the time. Therefore, we changed our manuscript (line 127): “The incidence of postoperative respiratory failure changed from year to year; however, a trend along the study period could not be detected.”

---

## [Editor Report · Decision Letter 1]

4 May 2020

PONE-D-19-32192R1

Postoperative acute respiratory dysfunction and the influence of antibiotics after acute type A aortic dissection surgery: a retrospective analysis

PLOS ONE

Dear Dr. Moeller,

Thank you for submitting your manuscript to PLOS ONE. After careful consideration, we have decided that your manuscript does not meet our criteria for publication and must therefore be rejected.

Specifically: the manuscript does not provide any new knowledge. 

I am sorry that we cannot be more positive on this occasion, but hope that you appreciate the reasons for this decision.

Yours sincerely,

Dong-Xin Wang

Academic Editor

PLOS ONE

Additional Editor Comments:

Acute respiratory dysfunction (ARD) is not a widely accepted diagnosis for postoperative pulmonary complications (see Anesthesiology 2010; 113:1338 –50). There are many postoperative pulmonary complications which are usually correlated with each other. For example, respiratory failure may be related to any other respiratory complications; it may also be related to cardiac failure. Furthermore, the severity of postoperative complications are different. A class II or higher on the Clavien-Dindo classification is usually accepted as a complication (see Ann Surg 2009;250: 177–186). The editor encourage the authors to make endpoint diagnosis according to widely accepted criteria. A composite endpoint may be better than a single diagnosis. They should then reanalyze data and resubmit their manuscript after revision. 

- - - - -

---

## [Author Response · Author response to Decision Letter 1]

3 Sep 2020

Dear editor and reviewers,

We would like to thank the academic editor Mrs. Balotta and the reviewers for their extensive and constructive review of our manuscript. 

We have revised our paper and hope that our careful revision and our respective statements are satisfying. The following is a detailed response to the editors’ and reviewers’ comments. 

Editor:

Journal Requirements:

=> Changes have been made in line 180, 300, 306, 313, 328, 342, 361, 381, 400, and 401.

=> PACE has been used before the first submission.

a) If there are ethical or legal restrictions on sharing a de-identified data set, please explain them in detail (e.g., data contain potentially identifyingor sensitive patient information) and who has imposed them (e.g., an ethics committee). Please also provide contact information for a data access committee, ethics committee, or other institutional body to which data requests may be sent.

=> As we pointed out, the data are legally restricted and cannot be shared publicly. They are available any time upon request by contacting our Department at the University of Regensburg. The head of our research laboratory, Dr. Michael Gruber takes care of a safe data backup.

=> https://www.ukr.de/kliniken-institute/Anaesthesiologie/Klinikteam___Mitarbeiter/Forschungslabor/index.php

Reviewer #1: Tahnk you for your paper.

The paper is interesting because it consider the impact of our antibiotic on the outcome of our patients, despite that in my opinion I think the use of broad spectrumantibiotic is influence non only by the lung status but also from the clinical analysis.

Patients who are going clinically worst need to be covered by broad spectrum non only because the lung problem.

I don't understand how patients who develop ARF can die less then who don't, it might be the selection of ARF definition.

=> Yes you are right, we were also surprised by our results. There might be a selection bias due to the ARD definition. However, other factors might indepently influence the outcome. For example, a recent meta-analysis showed that the body mass index can predict the clinical outcomes for patients with acute respiratory distress syndrome. Ni et al. showed that obesity and morbid obesity was associated with a lower mortality in patients with ARDS. We found a higher BMI in our ARD group. Accordingly, this might have influenced our outcome with a lower mortality. With regard to your very important annotation, we added the following sentence in line 218: “But it should not be neglected that it could also be a selection bias due to the ARD definition.” Furthermore, we revised our sentence in line 274: “Second, the different ARD definitions used in other studies make it difficult to compare their results with each other and with our results; and it might have had an impact on our results.” 

Reviewer #2: The Authors investigated the incidence, risk factors and impact on clinical outcomes of acute respiratory failure after surgery for acute type A aortic dissection. The retrospective cohort spans from 2008 to 2017.

I only have a few comments:

- did surgical techniques and cerebral protection techniques change during the study period?

=>This is a very important annotation. It might be comprehensible that over the time period of 9 years a change in the surgical technique and/or cerebral protection might have occurred. Unfortunately, this was not well documented. 

=> As we totally agree, we therefore changed the manuscript (line 282): “Fourth, we cannot clarify if the operation technique and cerebral protection has changed over the years. This might have had an impact on the outcome of the patients.”

- most patients were managed with moderate hypothermia. Did this change over time? If so, did this impact on transfusional requirements? Do the Authors believe that this had an impact on respiratory complications?

=>The management of hypothermia did not change over the time. Therefore, transfusional requirements were not different. These results are listed in table 2. To respond to this important annotation, we changed our manuscript (line 66): “The management of hypothermia did not change over the time.”

- did the incidence of postoperative respiratoryfailure change along the study period?

=> This is a very interesting question! We have looked at our data. We found a different incidence of postoperative respiratory failure per year. However, we did not see a trend over the time. Therefore, we changed our manuscript (line 127): “The incidence of postoperative respiratory failure changed from year to year; however, a trend along the study period could not be detected.”

---

## [Editor Report · Decision Letter 2]

1 Dec 2020

PONE-D-19-32192R2

Postoperative acute respiratory dysfunction and the influence of antibiotics after acute type A aortic dissection surgery: a retrospective analysis

PLOS ONE

Dear Dr. Möller,

Thank you for submitting your manuscript to PLOS ONE. After careful consideration, we feel that it has merit but does not fully meet PLOS ONE’s publication criteria as it currently stands. Therefore, we invite you to submit a revised version of the manuscript that addresses the points raised during the review process.

Please see below my comments and suggestions regarding the manuscript.

We look forward to receiving your revised manuscript.

Kind regards,

Aleksandar R. Zivkovic

Academic Editor

PLOS ONE

Journal Requirements:

1. Please provide additional details regarding participant consent. In the Methods section, please ensure that you have specified (1) whether consent was informed and (2) what type you obtained (for instance, written or verbal). If your study included minors, state whether you obtained consent from parents or guardians. If the need for consent was waived by the ethics committee, please include this information.

Additional Editor Comments (if provided):

The manuscript has merits to be accepted for the publication, however, I would like to point out the following matters:

Please consider addressing the following major points:

1. It is not clear which criteria were used to make the decision on using extended/broad spectrum antibiotics. Please describe the criteria. In case the criteria are not available/ the decision was made based on the attending clinician’s discretion, this, as well, needs to be described.

2. Please consider including only two patient groups: patients with acute respiratory dysfunction (ARD) and those without (NON-ARD). Furthermore, I would suggest describing the profile and the percentage of patients receiving narrow- or broad spectrum antibiotics within these two groups. Additional subdivision of the patients into the further two groups (based on narrow/broad spectrum antibiotics) could place the results out of focus. The main finding, that the type of the antibiotic regime did not correlate with the occurrence of the ARD, could be better emphasized if the authors focus on the two patient groups and describe the findings accordingly.

Minor points:

1. Line 23: please, introduce (define) the PF ratio: “PaO2/FiO2≤ 200 mmHg (PF ratio) within…”

2. Line 29: please, stay consistent in describing patient groups: first ARD, followed by the NON-ARD (or, alternatively, the other way around), but, please, keep consistency throughout the abstract text.

3. Line 51: please, consider rephrasing the term “certain”.

4. Line 61: Please, rephrase the following: “especially”

5. Line 64: please, remove “commonly”.

6. Line 66: please, consider rephrasing the following sentence: “For cerebral protection hypothermia was induced”

7. Line 77: please, remove “typical”

8. Line 78: please, consider rephrasing the following: “The main criteria for extubation were… the level of consciousness… the ability to follow instructions…” The criterion for the extubation is, presumably, the sufficient level of consciousness (RASS? GCS?), which includes the ability to follow instructions.

9. Line 104: please use “skewed distribution” instead of “skewed distributed”

10. Line 127: it is not clear what was significantly different. Please, rephrase this sentence and include the statistics. Consider pointing out whether the observed difference regards to the groups or the time points of the measurements.

11. Line 149: Please rephrase the term “slightly”

12. Line 172: please, check the sentence: “… than those in the patients treated with…”

13. Line 229: It is not clear whether authors refer to the findings of their study or the current literature regarding the term “at the moment”. Please rephrase.

14. Line 238: please consider rephrasing “… than those of patients who…”

15. Line: 248: “because this was not the focus of our study”. Please rephrase. The reason for not knowing which patients experienced malperfusion might be the fact that the records regarding this information are missing, however, not the fact that the it is “not the focus of the study”.

16. Line 264: please change “did not led” into “did not lead”

17. Line 266: please change “assumingly” into “presumably”. Moreover, consider discussing the option that patients who received broad spectrum antibiotics showed clinically more severe illness than those receiving narrow spectrum antibiotics. Do authors have access to disease severity scores obtained at the ICU (SOFA, APACHE, SAPS, TISS)?

18. Lines 266 and 268: please, consider rephrasing the following: can antibiotic therapy be “more effective”/ “less intensified”?

19. Line 269: please consider rephrasing the first sentence.

20. Line 274: consider using the term: multi-centre study.

21. Line 286: please check: “than that in patients”

22. Line 287: please, remove/change the term “appear”

23. Line 289: please check: “than those in patients”
---

## [Author Response · Author response to Decision Letter 2]

23 Jan 2021

Dear editor,

We would like to thank the academic editor Mr. Zivkovic for his extensive and constructive review of our manuscript. We have not found any comments from reviewers and are therefore only guided by the comments from the Academic editor.

We have revised our paper and hope that our careful revision and our respective statements are satisfying. The following is a detailed response to the editors’ comments. Our line specifications refer to the “revised manuscript with track changes”.

Editor:

Journal Requirements:

1. Please provide additional details regarding participant consent. In the Methods section, please ensure that you have specified (1) whether consent was informed and (2) what type you obtained (for instance, written or verbal). If your study included minors, state whether youobtained consent from parents or guardians. If the need for consent was waived by the ethics committee, please include this information.

-> With regard to your very important annotation, we added the following sentence in line 62: “Due to the design of the study an informed consent was not necessary.”

-> The Ethics Statement field of the submission form has been changed.

Additional Editor Comments (if provided):

The manuscript has merits to be accepted for the publication, however, I would like to point out the following matters:

Please consider addressing the following major points:

1. It is not clear which criteria were used to make the decision on using extended/broad spectrum antibiotics. Please describe the criteria. In case the criteria are not available/ the decision was made based on the attending clinician’s discretion, this, as well, needs to be described.

-> Yes, the decision was made by the physician in charge. Line 90: “Patients received either narrow-spectrum or broad-spectrum antibiotics depending on the physician in charge.”

To make the sentence more prominent, it was moved to the end of the section “Anesthetic and ICU management”. 

2. Please consider including only two patient groups: patients with acute respiratory dysfunction (ARD) and those without (NON-ARD). Furthermore, I would suggest describing the profile and the percentage of patients receiving narrow- or broad spectrum antibiotics within these two groups. Additional subdivision of the patients into the further two groups (based on narrow/broad spectrum antibiotics) could place the results out of focus. The main finding, that the type of the antibiotic regime did not correlate with the occurrence of the ARD, could be better emphasized if the authors focus on the two patient groups and describe the findings accordingly.

-> Thank you for the valuable comment. We completely agree with you and have therefore made some changes to focus more on the groups NON-ARD and ARD. 

Line 24: “All patients received either narrow- or broad-spectrum antibiotics.”

Line 32: “The incidence of pneumonia was independent of the antibiotic treatment regime (p = .391). Renal and neurological complication rate was higher in patients treated with broad-spectrum antibiotic.”

We have deleted the paragraph Classification of antibiotic groups and inserted therefore the sentence (line 90): “Patients received either narrow-spectrum or broad-spectrum antibiotics depending on the physician in charge. Narrow-spectrum antibiotic was normally a one day course of cefuroxime and broad-spectrum antibiotic was normally piperacillin/tazobactam for several days. Both were started within the first 24 hours after admission.”

Lines 167-179 have been rewritten: “In the NON-ARD group 43% of the patients (n = 39) were treated with a narrow-spectrum antibiotic and 57% (n = 51) […]”

Lines 259-263 and 267-271 have also been rewritten.

The sentence in line 324/325 has been deleted.

Supporting information S1 and S2 table has been removed.

Figure 3 has been changed into a new figur.

Minor points:

1. Line 23: please, introduce (define) the PF ratio: “PaO2/FiO2≤ 200 mmHg (PF ratio) within…”

-> The sentence has been corrected: “ARD was defined as PaO2/FiO2 ≤ 200 mmHg (PF ratio) within 48 hours after surgery.”

2. Line 29: please, stayconsistent in describing patient groups: first ARD, followed by the NON-ARD (or, alternatively, the other way around), but, please, keep consistency throughout the abstract text.

-> The abstract has been revised and changes have been made.

3. Line 51: please, consider rephrasing the term “certain”.

-> This has been changed (line 54).

4. Line 61: Please, rephrase the following: “especially”

-> “Especially” has been changed into “particularly” (line 65).

5. Line 64: please, remove “commonly”.

-> We removed the word “commonly” (line 68).

6. Line 66: please, consider rephrasing the following sentence: “For cerebral protection hypothermia was induced”

-> Thank you, the sentence has been rewritten: “Hypothermia was induced for cerebral protection.” (line 70).

7. Line 77: please, remove “typical”

-> The word “typical” has been removed (line 84).

8. Line 78: please, consider rephrasing the following: “The main criteria for extubation were… the level of consciousness… the ability to follow instructions…” The criterion for the extubation is, presumably, the sufficient level of consciousness (RASS? GCS?), which includes the ability to follow instructions.

-> You are absolutely right. In our intensive care unit GCS and RASS is used to determine the level of consciousness. The sentence has been corrected in line 84: “The main criteria for extubation were adequate gas exchange, the level of consciousness (for example GCS and RASS), normothermia, hemodynamic stability, a positive cough reflex and no significant bleeding.”

9. Line 104: please use “skewed distribution”instead of “skewed distributed”

-> This error has been corrected in line 118.

10. Line 127: it is not clear what was significantly different. Please, rephrase this sentence and include the statistics. Consider pointing out whether the observed difference regards to the groups or the time points of the measurements.

-> This is a very important annotation. Therefore, we changed our manuscript (line 140): “Each PF ratio between the ARD and NON-ARD groups was significantly different from the preoperative admission until the seventh postoperative day.”

11. Line 149: Please rephrase the term “slightly”

-> The word has been changed into “a few” (line 162).

12. Line 172: please, check the sentence: “… than those in the patients treated with…”

-> Sentences have been corrected (line 27, line 206, line 216, line 239 and line 245).

13. Line 229: It is not clear whether authors refer to the findings of their study or the current literature regarding the term “at the moment”. Please rephrase.

-> The phrase “in the literature” has been added (line 259).

14. Line 238: please consider rephrasing “… than those of patients who…”

-> The sentence in line 271 has been removed. The sentence in line 302 has been corrected.

15. Line: 248: “because this was not the focus of our study”. Please rephrase. The reason for not knowing which patients experienced malperfusion might be the fact that the records regarding this information aremissing, however, not the fact that the it is “not the focus of the study”.

-> This is absolutely correct. The sentence has been corrected into (line 282): “(…) because this was not documented.” 

16. Line 264: please change “did not led” into “did not lead”

-> This error has been corrected (line 298).

17. Line 266: please change “assumingly” into “presumably”. Moreover, consider discussing the option that patients who received broad spectrum antibiotics showed clinically more severe illness than those receiving narrow spectrum antibiotics. Do authors have access to disease severity scores obtained at the ICU (SOFA, APACHE, SAPS, TISS)?

-> “Assumingly” has been changed into “presumably” (line 300).

Unfortunately this was not sufficiently documented.

18. Lines 266 and 268: please, consider rephrasing the following: can antibiotic therapy be “more effective”/ “less intensified”?

-> Thank you. We therefore changed the manuscript (line 299): “Similarly, patients in the present study who received a broad-spectrum, and presumably more effective, antibiotic therapy did not have better postoperative outcomes and had more nephrological and neurological complications than patients who received a narrow-spectrum antibiotic treatment.”

19. Line 269: please consider rephrasing the first sentence.

-> The sentence has been rewritten (line 304): ”The present study has some potential limitations due to its retrospective design.”

20. Line 274: consider using the term: multi-centre study.

-> The word “multi-centre study” has been used (line 309).

21. Line 286: please check: “than that in patients”

-> This has been corrected (line 323).

22. Line 287: please, remove/change the term “appear”

-> The sentence has been deleted (line 325).

23. Line 289: please check: “than those in patients”

-> Thank you. This has been corrected (line 326).

Reviewers' comments:

 /

---

## [Editor Report · Decision Letter 3]

26 Jan 2021

Postoperative acute respiratory dysfunction and the influence of antibiotics after acute type A aortic dissection surgery: a retrospective analysis

PONE-D-19-32192R3

Dear Dr. Möller,

We’re pleased to inform you that your manuscript has been judged scientifically suitable for publication and will be formally accepted for publication once it meets all outstanding technical requirements.

Kind regards,

Aleksandar R. Zivkovic

Academic Editor

PLOS ONE

---

## [Editor Report · Acceptance letter]

1 Feb 2021

PONE-D-19-32192R3 

Postoperative acute respiratory dysfunction and the influence of antibiotics after acute type A aortic dissection surgery: a retrospective analysis 

Dear Dr. Möller:

I'm pleased to inform you that your manuscript has been deemed suitable for publication in PLOS ONE. Congratulations! Your manuscript is now with our production department. 

Kind regards, 

on behalf of

Dr. Aleksandar R. Zivkovic 

Academic Editor

PLOS ONE